# Hypoxia-Targeted Immunotherapy with PD-1 Blockade in Head and Neck Cancer

**DOI:** 10.3390/cancers16173013

**Published:** 2024-08-29

**Authors:** Risa Wakisaka, Hidekiyo Yamaki, Michihisa Kono, Takahiro Inoue, Ryosuke Sato, Hiroki Komatsuda, Kenzo Ohara, Akemi Kosaka, Takayuki Ohkuri, Toshihiro Nagato, Kan Kishibe, Koh Nakayama, Hiroya Kobayashi, Takumi Kumai, Miki Takahara

**Affiliations:** 1Department of Otolaryngology-Head and Neck Surgery, Asahikawa Medical University, Asahikawa 0788510, Japanrsato@asahikawa-med.ac.jp (R.S.); komatsuda@asahikawa-med.ac.jp (H.K.);; 2Department of Innovative Head & Neck Cancer Research and Treatment (IHNCRT), Asahikawa Medical University, Asahikawa 0788510, Japan; 3Department of Pathology, Asahikawa Medical University, Asahikawa 0788510, Japan; kosakaa@asahikawa-med.ac.jp (A.K.); ohkurit@asahikawa-med.ac.jp (T.O.); rijun@asahikawa-med.ac.jp (T.N.);; 4Department of Pharmacology, Asahikawa Medical University, Asahikawa 0788510, Japan; knakayama@asahikawa-med.ac.jp

**Keywords:** hypoxia, MutT homolog-1, immunotherapy, head and neck cancer

## Abstract

**Simple Summary:**

Intratumoral hypoxia is associated with poor prognosis and therapeutic resistance by modifying the tumor microenvironment in several cancers. MTH1, a member of the Nudix family, maintains the genomic integrity and viability of tumor cells under hypoxic conditions. This study aimed to investigate whether the antitumor activity of immune cells is effective under hypoxia and whether hypoxia-induced MTH1 could be a target for immunotherapy. We found that MTH1 expression is elevated in hypoxic head and neck cancer cell lines and tissues. A novel MTH1 epitope peptide activates CD4+ helper T cells with cytotoxic activity, and is effective even under hypoxic conditions. Combining MTH1-targeted immunotherapy with PD-1 blockade enhances cytotoxicity. These results suggest that MTH1-targeted immunotherapy combined with checkpoint blockade could effectively treat hypoxic tumors by maintaining T-cell activity and increasing cytotoxicity under hypoxic conditions.

**Abstract:**

Intratumoral hypoxia is associated with tumor progression, aggressiveness, and therapeutic resistance in several cancers. Hypoxia causes cancer cells to experience replication stress, thereby activating DNA damage and repair pathways. MutT homologue-1 (MTH1, also known as NUDT1), a member of the Nudix family, maintains the genomic integrity and viability of tumor cells in the hypoxic tumor microenvironment. Although hypoxia is associated with poor prognosis and can cause therapeutic resistance by regulating the microenvironment, it has not been considered a treatable target in cancer. This study aimed to investigate whether hypoxia-induced MTH1 is a useful target for immunotherapy and whether hypoxic conditions influence the antitumor activity of immune cells. Our results showed that MTH1 expression was elevated under hypoxic conditions in head and neck cancer cell lines. Furthermore, we identified a novel MTH1-targeting epitope peptide that can activate peptide-specific CD4+ helper T cells with cytotoxic activity. The proliferation and cytotoxic activity of T cells were maintained under hypoxic conditions, and PD-1 blockade further augmented the cytotoxicity. These results indicate that MTH1-targeted immunotherapy combined with checkpoint blockade can be an effective strategy for the treatment of hypoxic tumors.

## 1. Introduction

Head and neck squamous cell carcinoma (HNSCC) is the seventh most common type of cancer worldwide with >60% of patients having advanced-stage cancer [1]. According to Global Cancer Statistics (GLOBCAN), HNSCC accounted for approximately 3% of all human cancers (51,540 new cases) in 2018 [2,3]. Although a combination of surgery, chemotherapy, and radiotherapy is considered the standard therapy for HNSCC, treatment resistance is commonly observed in advanced stages. The risk of local recurrence in advanced HNSCC is 15–40%, and distant metastasis results in a 5-year overall survival rate of <50% [4].

Hypoxia is a distinguishing feature of the tumor microenvironment (TME) that is present in most solid tumors [5]. Approximately 50–60% of locally advanced solid tumors have a hypoxic area caused by an imbalance between increased oxygen consumption and insufficient oxygen delivery [6,7]. Intratumoral hypoxia is associated with tumor progression, aggressiveness, and therapeutic resistance in several cancers, including head and neck cancer [8]. Hypoxia-inducible factor 1 (HIF-1) controls oxygen delivery and metabolic adaptation to hypoxia via changes in tissue oxygenation [9,10]. HIF-1 consists of α and β subunits which are involved in the adaptive response of cells to hypoxia. Genes regulated by HIF-1 include proliferation factors, anaerobic glycolytic enzymes, and others involved in tumor growth, survival, invasion, metastasis, immune activation, and treatment response [11,12,13]. Overexpression of HIF-1 triggers the transcription of vascular endothelial growth factor, which stimulates angiogenesis and contributes to tumor progression and metastasis [11,14,15].

Hypoxia causes cancer cells to experience replication stress, thereby activating DNA damage and repair pathways [6]. MutT homologue-1 (MTH1), also called NUDT1, is a member of the Nudix family that maintains the genomic integrity and viability of tumor cells [16]. MTH1 is activated by increased levels of reactive oxygen species (ROS) in cancer cells. Furthermore, MTH1 hydrolyzes oxidized nucleoside triphosphates such as 8-oxodGTP or 2-OH-dATP, thereby hydrolyzing their corresponding monophosphates to prevent DNA damage and protect against cancer cell death [17,18,19]. Overexpression of MTH1 is associated with poor clinical staging in non-small cell lung carcinoma and breast cancer [20,21]. Qiu et al. demonstrated that MTH1 expression was upregulated via HIF-1 in response to hypoxic stress in human colorectal cancer cells [22], suggesting that MTH1 may be a useful molecular marker of hypoxia. However, the expression of MTH1 and its therapeutic use in HNSCC remain unknown.

Because hypoxia is associated with poor prognosis by inducing therapeutic resistance through modification of the TME, hypoxia-targeting immunotherapy can be an interesting approach for treating otherwise untreatable tumors. This study aimed to investigate immunological changes in tumors and T cells under hypoxic conditions. We also examined whether MTH1 is a useful target for hypoxia-targeted immunotherapy.

## 2. Materials and Methods

### 2.1. Cell Lines

The following HNSCC cell lines were used in this study: SAS (tongue SCC; HLA-DR 9, 15, and 53), Sa-3 (human gingival SCC; HLA-DR9, 10, and 53), HSC2 (human oral SCC; HLA-DR13), and HSC4 (human tongue SCC; HLA-DR1, 4, and 53). HSC2 and HSC4 cells were supplied by the RIKEN BioResource Center (Tsukuba, Ibaraki, Japan). SAS cells (human tongue SCC) were purchased from the American Type Culture Collection (Manassas, VA, USA). Normal human bronchial epithelial cells (NHBE) were kindly provided by Dr. Hirohito Kita (Mayo clinic). All cell lines were maintained in tissue culture as per the supplier’s recommendations.

### 2.2. Patients and Immunohistochemistry (IHC)

Pre-treatment biopsy tissues were obtained from the primary sites of 55 patients with oropharyngeal cancer treated at Asahikawa Medical University. Informed consent was obtained from the Asahikawa Medical University website using the opt-out method. The collection and analysis of clinical data were approved by the Asahikawa Medical University Ethical Committee and Review Board (#20054 and #23070). The clinical characteristics of the patients are summarized in Table 1.

TNM staging was based on the 8th edition of the International Union Against Cancer Staging System. MTH1 expression was analyzed in formalin-fixed and paraffin-embedded tissues (FFPE) from patients with HNSCC. For IHC, a rabbit polyclonal antibody (Ab) against MTH1 (ab187531; 1:250 dilution; Abcam, Cambridge, MA, USA) was used as the primary Ab. The Ventana Benchmark GX and Ventana UltraView Universal DAB Detection Kit (Roche Diagnostics, Penzberg, Germany) were used. The staining intensity scores for tumoral MTH1 were graded as follows: 0, no staining; 1, weak; 2, moderate and 3, strong staining. Quantity scores for MTH1 were assessed from the percentage of stained tumor and graded as follows: 0, <5%; 1, 5–25%; 2, 26–50%; and 3, >50%. The IHC score was calculated by combining the staining intensity and quantity scores; ≥5 points indicated high expression, whereas ≤4 points indicated low expression. The expression of MTH1 was also analyzed using the Human Protein Atlas database (https://www.proteinatlas.org/, accessed on 12 December 2023).

### 2.3. Cell Culture and Hypoxic Conditions

T cells were activated using PMA (20 ng/mL) and ionomycin (1 μg/mL) for 24 h. Hypoxic conditions were achieved using a BIONIX-3 hypoxic cell culture kit (Sugiyamagen, Tokyo, Japan) and a hypoxia workstation (Hirasawa Works, Tokyo, Japan), in which the hypoxic environment (1% O_2_, 94% N_2_, and 5% CO_2_) and temperature (37 °C) were kept constant. The oxygen levels in normoxic or hypoxic tumor tissues are >2% or 1–2%, respectively [5,23]. Therefore, we used 1% O_2_ as the model for hypoxic conditions.

### 2.4. Synthetic Peptide

The Immune Epitope Database Analysis Resource (RRID: SCR_013182, https://www.iedb.org/, accessed on 10 December 2023) and SYFPEITHI (http://www.syfpeithi.de/, accessed on 10 December 2023) were used to investigate HLA-DR-binding amino acid sequence of MTH-1. We selected the MTH1_45–59_ (SRLYTLVLVLQPQRV) peptide with the potential to bind to multiple HLA-DRs (DR1, 4, 7, 11, and 15). The MTH-1_45–59_ peptide was synthesized by Hokkaido System Science.

### 2.5. Flow Cytometry 

The expression of HLA class I and HLA-DR on the tumor cell lines was assessed by flow cytometry using anti-HLA class I Abs conjugated with FITC (G46-2.6, BD Pharmingen, Heidelberg, Germany), anti-HLA-DR Abs conjugated with PE (L243, BD Pharmingen, Germany), and anti-PD-L1 (MIH1, eBioscience, San Diego, CA, USA) Abs conjugated with PE. Furthermore, HNSCC cell lines were treated with or without 50 U/mL IFN-γ for 48 h prior to the assay. Intracellular MTH1 and HIF-1α staining was performed using Perm/WashTM and Cytofix/CytopermTM (BD Pharmingen, Germany), with anti-MTH1 (ab187531, Abcam) and anti-HIF-1α mAbs (ab51608, Abcam, Cambridge, UK) as the primary Abs, and PE donkey anti-rabbit Ab (eBioscience, San Diego, CA, USA) as the secondary Ab. Samples were analyzed using a CytoFLEX LX flow cytometer and CytExpert (Beckman Coulter, Brea, CA, USA).

### 2.6. In Vitro Induction of MTH1-Specific CD4+ T Cells

Peptide-reactive CD4+ T cell (HTL) lines were prepared from healthy donor peripheral blood mononuclear cells (PBMCs) as previously described [24]. Briefly, monocytes and HTLs were purified from PBMCs using the EasySepTM Human CD14+ Positive Selection Kit II and EasySepTM Human CD4+ T Cell Isolation Kit (STEMCELL Technology, Vancouver, Canada), respectively. Monocytes were differentiated into dendritic cells (DCs) using granulocyte–macrophage colony-stimulating factor (GM-CSF) (50 ng/mL, PeproTech, Rocky Hill, CT, USA) and interleukin-4 (IL-4) (1000 IU/mL, PeproTech, Rocky Hill, NJ, USA) for 7 days. The HTLs were stimulated with MTH1_45–59_ peptide-pulsed autologous DCs for a cycle and 40Gy γ-irradiated autologous PBMCs for 2 cycles. The HTLs were assessed for IFN-γ production with MTH1_45–59_ peptide stimulation (IFN-γ ELISA kits, BD Pharmingen, Germany). Microcultures with IFN-γ production after peptide stimulation were isolated by limiting dilution. AIM-V medium (Thermo Fisher, Waltham, MA, USA) supplemented with 3% human male AB serum (Innovative Research, Novi, MI, USA) was used as the complete culture medium for all the experiments.

### 2.7. Antigen-Specific Responses by MTH1-Specific HTLs

The responses of the HTLs to the peptides were assessed as previously described [25]. Irradiated autologous PBMCs or MTH1-expressing HNSCC cell lines were used as antigen-presenting cells (APCs). HLA restriction was evaluated by anti-HLA-DR Ab L243 and anti-HLA class I Ab W6/32. To upregulate HLA-DR expression, HNSCC cells were treated with IFN-γ (500 U/mL; PeproTech, Rocky Hill, CT, USA) for 48 h.

### 2.8. Cytotoxicity Assay

Supernatants of MTH1-specific HTLs co-cultured with tumor cell lines were assessed using Human IFN-γ ELISA kits (R&D Systems, Minneapolis, MN, USA). Direct killing activity was assessed by labeling target tumor cell lines using the CellTraceTM CFSE Cell Proliferation Kit (Invitrogen, Carlsbad, CA, USA). After 6 h of co-culture with various effector/target cell (E:T) ratios of MTH1-specific HTLs and tumors, the number of dead tumor cells (CFSE+) labeled using a 7-AAD viability staining solution (BioLegend, SanDiego, CA, USA) was quantified by flow cytometry.

### 2.9. Cell Cycle Assay

Cell cycle analysis was performed using the Cell Cycle Assay Solution Deep Red Kit (Dojindo, Kumamoto, Japan). Cells were incubated for 15 min at 37 °C, according to the manufacturer’s instructions. The stained cells were analyzed using flow cytometry.

### 2.10. Statistical Analysis

All data were assessed using the Student’s *t*- or Fisher’s exact tests. All results are presented as mean ± SD. Statistical analysis was performed using the Student’s *t*-test and one- or two-way ANOVA. Statistical significance was set at * *p* < 0.05, ** *p* < 0.01, *** *p* < 0.001, and **** *p* < 0.0001. *p*-values were calculated using GraphPad Prism 9 (GraphPad Software, Inc. ver. 9.4.1).

## 3. Results

### 3.1. Immunogenic Changes in HNSCC Cell Lines under Hypoxic Condition

A hypoxic environment exists in solid tumors owing to an imbalance between oxygen supply and demand. Because oxygen concentrations in tumors have been reported to range from 1% to 2% [23], 1% was chosen as a condition that represents hypoxic tumor in this study. To develop a hypoxia-targeted immunotherapy, we first examined the antigen-presenting machinery, including HLA-DR and HLA-class I, in HNSCC cell lines (SAS, Sa-3, HSC2, and HSC4) under hypoxic conditions. As shown in Figure 1A,B, HLA-DR and HLA-class I were upregulated by IFN-γ stimulation and this upregulation was maintained under hypoxic conditions. Notably, IFN-γ-induced HLA-DR levels significantly increased after 24 h hypoxia treatment compared to under normoxic conditions. The expression of HLA class I under hypoxic and normoxic conditions was comparable in two cell lines (SAS and Sa-3), upregulated in HSC4 cells, and downregulated in HSC2 cells. Next, we examined the expression of the negative immune checkpoint, PD-L1. PD-L1 levels were upregulated in three HNSCC cell lines (SAS, Sa-3, and HSC4) without IFN-γ under hypoxic conditions (Figure 1C).

### 3.2. Viability and Proliferation of T Cells under Hypoxic Conditions

Proliferation of antitumor T cells is necessary for the development of T-cell-based immunotherapy. To determine whether immunotherapy is effective under hypoxic conditions, the viability and mitogenicity of immune cells were investigated under hypoxic conditions. The proliferation ratios of CD4+ and CD8+ T cells after 48 h of culture under hypoxic conditions, with or without PMA/ionomycin stimulation, are shown in Figure 2A. CD4+ and CD8+ T cells retained their proliferative activity with PMA/ionomycin stimulation under hypoxic conditions.

Next, cell cycle analysis was performed using propidium iodide staining and flow cytometry under hypoxic conditions with or without PMA/ionomycin stimulation for 24 h. As shown in Figure 2B,C, the percentage of cells that entered into G2/M phase was augmented by stimulation under hypoxic conditions. Although CD4 T cells in the G2/M phase were increased in normoxia compared to hypoxia, the proliferation of CD4 T cells was comparable regardless of oxide concentration (Appendix A). These results suggest that stimulated T cells maintain their proliferative and mitogenic potential under hypoxia, at least in this short-term assay.

### 3.3. MTH1 Expression in Patients with HNSCC and HNSCC Cell Lines

The identification of an epitope from a relevant antigen is required to activate antigen-specific T cells. The finding that T cells can proliferate under hypoxic conditions has led us to focus on proteins upregulated under hypoxia as targets for immunotherapy. Although MTH1 is upregulated under hypoxic conditions in colon cancer cell lines [22], its induction under hypoxic conditions in head and neck cancer remains unknown. To evaluate whether MTH1 could be induced by hypoxia, we first confirmed that 1% O_2_ could induce hypoxic responses in HNSCC cell lines. As shown in Appendix A, 1% O_2_ could elicit the expression of HIF-1α, a representative hypoxia marker, in these cell lines. Next, we examined MTH1 expression in HNSCC cell lines. As shown in Figure 3A, MTH1 was expressed in all examined HNSCC cell lines. Notably, MTH1 expression in SAS, HSC2, and HSC4 cells was upregulated under hypoxic conditions. The average values of mean fluorescence intensity (MFI) are shown in Appendix A. MTH1 expression remained comparable with IFN-γ stimulation (Appendix A). The expression of MTH1 was increased in tumor compared to normal cells (Appendix A). Because the MTH1 dependency is low in normal tissue compared to tumor tissue [26], hypoxia did not increase MTH1 expression in normal cells (Appendix A).

Next, we performed IHC staining of tissues from 55 patients with oropharyngeal cancer, and evaluated the correlation between MTH1 expression and clinical characteristics. As a result, MTH1 was expressed in 39/55 (70.9%) of HNSCC tissue samples (Figure 3B,C). No correlation was observed between MTH1 expression and sex, age, HPV status, or tumor stage (Table 1). Comparing overall survival and disease-free survival according to MTH1 expression intensity, patients with high MTH1 expression tended to have worse disease-free survival, but not overall survival (Appendix A). Database analysis using the Human Protein Atlas also showed that patients with MTH1-high HNSCC tended to have a worse prognosis (Figure 3D). These results showed that MTH1 is expressed in most HNSCC and is elevated under hypoxic conditions, suggesting that this protein could be a potential target for hypoxia-targeted immunotherapy.

### 3.4. Generation of MTH1_45–59_ Reactive CD4+ T-Cell Lines with Cytotoxic Activity

Based on the finding that HNSCC cells expressed HLA-DR under hypoxic conditions (Figure 1A), we focused on HLA class II-restricted epitopes that elicit HTL responses. The MTH1_45–59_ peptide that can bind to multiple HLA-DRs was identified using a computer-based algorithm as a potential candidate epitope for inducing CD4+ HTL responses. MTH1_45–59_-reactive HTLs were established from a healthy donor with HLA-DR 9/12/53. The MTH1-reactive HTLs produced IFN-γ in response to MTH1 peptides (Figure 4A). To confirm HLA restriction, MTH1-reactive HTLs were cultured with peptide-pulsed autologous PBMCs with an anti-HLA-DR mAb. As shown in Figure 4B, the response of HTLs to MTH1-derived peptides was significantly suppressed by an anti-HLA-DR mAb indicating that T cells recognize peptides present on HLA-DR.

Direct recognition of MTH1_45–59_-reactive HTLs in HNSCC cells was examined in a co-culture system. These HTLs produced IFN-γ in response to HLA-DR53+ but not HLA-unmatched HNSCC cells (Figure 4C). In addition to IFN-γ, granzyme B, a cytolytic protease was also produced by these T cells which recognized HLA-DR53+ HNSCC cells (Figure 4D), indicating that MTH1-induced CD4+ HTL lines are cytotoxic CD4+ HTLs. To confirm whether MTH1_45–59_-reactive CD4+ HTLs exhibit direct tumor cytotoxicity, CFSE-labeled tumors were co-cultured with these T cells. As shown in Figure 4E,F, MTH1_45–59_-reactive CD4+ HTLs directly killed HLA-DR-matched tumors. Taken together, the novel MTH1_45–59_ peptide can be a potent epitope for eliciting antitumor HTL responses.

### 3.5. MTH1_45–59_-Reactive T Cells in Periphery Blood from Patients with HNSCC

To translate our findings into peptide vaccines in a clinical setting, the presence of MTH1-reactive precursor T cells in patients with HNSCC is mandatory. Therefore, we evaluated the presence of MTH1-reactive T cells by stimulating PBMCs isolated from four patients with HNSCC with the MTH1_45–59_ peptide for two cycles every 7 days, and IFN-γ production was measured. As shown in Figure 5A,B, T cells from patients with HNSCC responded to the MTH1_45–59_ peptide. In contrast, MTH1-reactive precursor T cells could not be detected in PBMCs from healthy donors using the short-term activation protocol (Figure 5C,D), indicating that the precursors of MTH1-reactive HTLs are more abundant in patients with HNSCC than in healthy donors.

### 3.6. Antitumor Activity of MTH1_45–59_-Reactive CD4+ T Cells with PD-1 Blockade in Hypoxia

To confirm the antitumor activity of MTH1_45–59_-reactive HTLs, we co-cultured T cells with HNSCC cell lines under hypoxic conditions. As shown in Figure 6A, T-cell responses against HNSCC cells were comparable to those under hypoxia and normoxia. Since PD-L1 expression in HNSCC cells was upregulated under hypoxic conditions (Figure 1C), an anti-PD-1 Ab was added to the T-cell-tumor co-culture system. Interestingly, PD-1 blockade significantly augmented IFN-γ production by T cells only under hypoxic conditions (Figure 6A). Although the cytotoxicity of MTH1_45–59_-reactive HTLs was comparable between hypoxic and normoxic conditions (Figure 6B), PD-1 blockade further upregulated cytotoxicity against HNSCC cells under hypoxic conditions than those under normoxic conditions. These results indicate that the immune adjuvant activity of checkpoint blockade could be remarkable for hypoxic tumor-targeting immunotherapy.

## 4. Discussion

An imbalance between oxygen supply and demand creates a hypoxic environment in solid tumors [27]. Although hypoxia is a hallmark of aggressive cancer, few studies have investigated therapies targeting hypoxia. This study revealed the potential of MTH1 as a hypoxia-targeting antigen for peptide vaccines combined with immune checkpoint inhibitors (ICIs). Antigen presentation in HNSCC cell lines was maintained even under hypoxic conditions, and a greater increase in HLA-DR (MHC class II) expression was observed under hypoxic conditions. Notably, MHC class II expression is regulated by class II transactivator (CIITA). Abu-El-Rub et al. reported that exposure to hypoxia upregulates MHC class II via Sug1 and CIITA in mesenchymal stem cells [28]. Because our group has shown that MHC class II in HNSCC is also regulated by CIITA [29], it is plausible that hypoxia induces HLA-DR through CIITA upregulation in HNSCC. Unlike HLA-DR, HLA class Ⅰ expression was lower under hypoxic conditions than under normoxia. Tumor cell shedding of MHC class I chain-related molecules through nitric oxide signaling in hypoxia may be responsible for this downregulation [30]. Other studies have reported that hypoxia induces tumor PD-L1 expression via HIF-1 but not HIF-2 [31,32]. These results suggest that HTLs that recognize peptides on MHC class II, but not CTLs, are promising effectors in hypoxia-targeted immunotherapy, and that checkpoint blockade can be a suitable immune adjuvant.

A few studies on CTL epitopes that target hypoxia have been reported [33,34]; however, an HTL epitope targeting hypoxia in cancer vaccines has not been detected. In this study, we identified a novel T-cell epitope peptide from MTH1 that induces antitumor HTL responses against MTH1-expressing tumors under hypoxic conditions. Since we showed that the MTH1_45–59_ peptide reacts with HLA-DR53-expressing tumors, our in silico sequencing results suggest that the MTH1_45–59_ peptide can bind to multiple common HLA-DR alleles to cover a broad population of patients (DRB1*0101, DRB1*0401, DRB1*0701, DRB1*1101, and DRB1*1501). Precursors of MTH1-reactive HTLs were identified in patients with HNSCC. These T cells also exist in healthy donors, but several cycles of stimulation by professional APCs were required to be detected due to the low number of cells compared to the patients. Although these T cells may have low-affinity T-cell receptors that pass through negative selection in the thymus, they can directly recognize and kill HNSCC cells, as shown in this study. Because the patients with MTH1-reactive precursor T cells have shown no autoimmune diseases, it is plausible that MTH1-reactive T cells may only react to MTH1-high tumor but ignore MTH1-low normal tissues.

Since CTLs are considered the main effectors in T cell-based immunotherapy, effective antitumor responses require both antigen-reactive CTLs and HTLs [35]. In addition to their direct cytotoxic effects [36], antigen-reactive HTLs activate CTLs, NK cells, and macrophages via the CD40L/CD40 pathway, and establish long-term antitumor memory [37,38]. Under hypoxic conditions, anaerobic glycolysis produces lactate, which may affect T-cell activation, proliferation, and cytotoxicity [39]. Gropper et al. reported that CTLs cultured with 1% O_2_ survive and mature well; however, their proliferation rate is slower than that with 21% O_2_ [40]. In this study, HTL proliferation was preserved under hypoxic conditions for a short incubation period. An in vivo model of hypoxic tumors treated with HTLs or a CTL-targeting peptide vaccine [41] is required to further examine the effect of chronic, but not transient, hypoxia on the activity of T cells.

Cancer cells accelerate metabolism with higher levels of ROS than normal cells, and MTH1 is essential for the survival of cancer cells by remediating DNA damage; however, it is not necessary for the proliferation of normal cells [19,26]. MTH1 is overexpressed in various cancers, including colorectal [42] and non-small cell lung cancer [43]. We found that MTH1 was expressed in approximately 70% of the patients with oropharyngeal cancer, and that patients with high MTH1 expression tended to have poor prognosis. Furthermore, MTH1 expression was highly localized in the tumor but not in the surrounding tissues. In addition to our study and the Human Protein Atlas database, Shen et al. reported that immunohistochemical detection of MTH1 was associated with poor prognosis and a pathological grade of HNSCC [44]. Qiu et al. reported that colon cancer cells increased MTH-1 mRNA and protein levels under hypoxic conditions via HIF-1 [22]. We also showed that MTH1 expression was upregulated under hypoxic conditions in HNSCC, suggesting that MTH1 may be a useful therapeutic target for hypoxic HNSCC with aggressive or treatment-resistant characteristics.

The low number of responders to immunotherapy remains a major challenge. In colorectal cancer, the HIF-1/PD-L1 pathway activated in hypoxia promotes immune escape [45]. Ivraym et al. reported that the exposure of human or murine cancer cells to hypoxia for 24 h upregulated PD-L1 in an HIF-1-dependent manner [46], indicating that PD-1/PD-L1 would be responsible for immune evasion under hypoxic conditions. We showed that the antitumor effect of MTH1-specific HTLs was enhanced by a combination with an anti-PD-1 Ab under hypoxic conditions, suggesting that the combination of peptide vaccines with ICIs could be a potent immunotherapeutic approach against aggressive tumors under hypoxic conditions. The limitation of hypoxia-targeting immunotherapy is the difficulty of T cells in accessing poorly perfused regions. It is possible that the vasodilation by nitric oxide in hypoxia [47] may increase the access of T cells to tumor. Since nitric oxide induces indoleamine 2,3-dioxygenase-1, an immune-suppressing enzyme [48,49], by heme allocation in hypoxia [50], the influence of nitric oxide against antitumor immunity remains to be determined. Further investigations including a combination of hypoxia-targeting peptide vaccines with poly-ICLC, which induces T-cell-recruiting chemokines from vascular endothelial cells [51], are required to ensure the recruitment of MTH1-targeted T cells into hypoxic tumors. Radiotherapy is a promising partner of immunotherapy by killing tumor cells followed by antigen spreading. The antitumor effect of radiotherapy partly relies on oxygen by inducing reactive oxygen species that damage tumoral DNA. The combination of radiotherapy with hypoxia-targeted immunotherapy would be an interesting strategy to treat tumors with both normoxic and hypoxic areas.

## 5. Conclusions

In this study, MTH1 expression was elevated under hypoxic conditions in HNSCC cells. We identified an MTH1-targeting epitope peptide that can induce peptide-specific HTLs with cytotoxic activity. The proliferation and cytotoxic activity of the T cells were maintained under hypoxic conditions, and their combination with an anti-PD-1 Ab further enhanced their cytotoxic activity (Figure 7), indicating that immunotherapy may be an effective strategy for targeting hypoxic tumors.

## Figures and Tables

**Figure 1 cancers-16-03013-f001:**
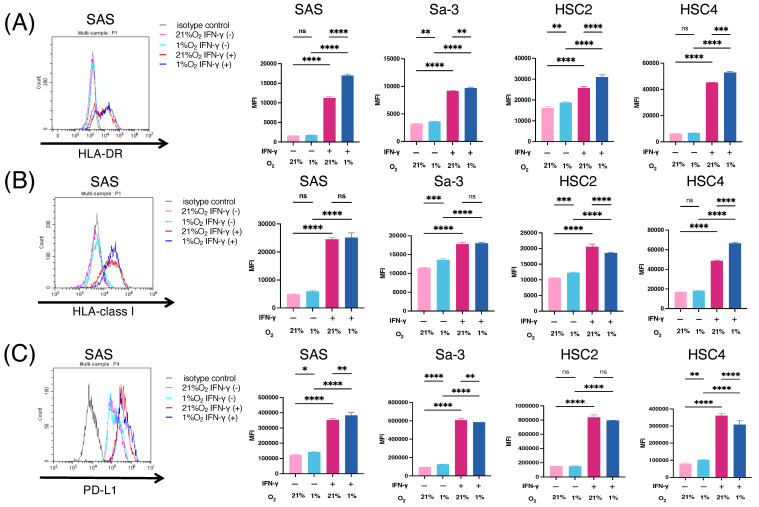
Antigen presenting molecules on HNSCC cell lines in hypoxic condition. (**A**–**C**) The expression levels of (**A**) HLA-DR, (**B**) HLA-class I, and (**C**) PD-L1 on HNSCC cell lines incubated with or without IFN-γ (50 IU/mL) were evaluated by flow cytometry. Black, isotype control; pink, tumor cells under normoxic conditions; light blue, tumor cells under hypoxic conditions; red, tumor cells treated with IFN-γ under normoxic conditions; blue, tumor cells treated with IFN-γ under hypoxic conditions. Experiments were performed in triplicate. Bars and error bars represent the mean and SD, respectively (* *p* < 0.05, ** *p* < 0.01, *** *p* < 0.001, **** *p* < 0.0001, one-way ANOVA).

**Figure 2 cancers-16-03013-f002:**
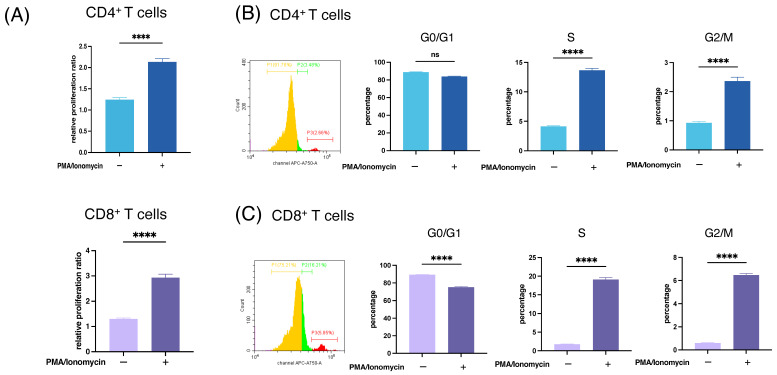
Viability and proliferation of T cells under hypoxic conditions. (**A**) Cell proliferation ratio of CD4+ and CD8+ T cells was assessed using the CCK-8 assay under hypoxic (1% O_2_) conditions with or without PMA/Ionomycin stimulation for 48 h. (**B**,**C**) Cell cycle analysis of CD4+ T cell and CD8+ T cell by propidium iodide staining and flow cytometry under hypoxic (1% O_2_) conditions, with or without PMA/Ionomycin stimulation for 24 h. Experiments were performed in triplicate. Bars and error bars represent the mean and SD, respectively (**** *p* < 0.0001, one-way ANOVA).

**Figure 3 cancers-16-03013-f003:**
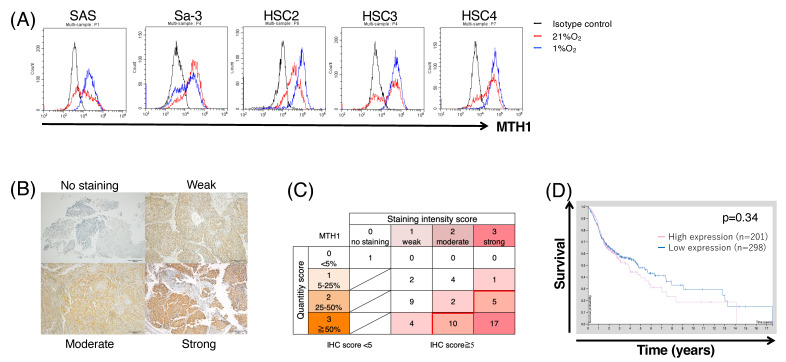
Expression of MTH1 in HNSCC patients and cell lines. (**A**) Intracellular staining of MTH1 in HNSCC cell lines. Secondary antibody with isotype control was used as a negative control. (**B**) Representative immunohistochemical (IHC) images of MTH1 in oropharyngeal cancer. Scale bar = 100 μm. (**C**) Distribution of IHC scores for MTH1. The IHC score was calculated as the sum of the staining intensities. IHC scores ≥ 5 were defined as high expression, and scores ≤ 4 were defined as low expression. (**D**) Expression of MTH1 was analyzed using the Human Protein Atlas database (https://www.proteinatlas.org/ENSG00000106268-NUDT1/pathology/head+and+neck+cancer, accessed on 12 December 2023). The 5-year survival rate tended to be lower in HNSCC patients with high MTH1 expression (cut-off value, 14.35; *p* score, 0.34).

**Figure 4 cancers-16-03013-f004:**
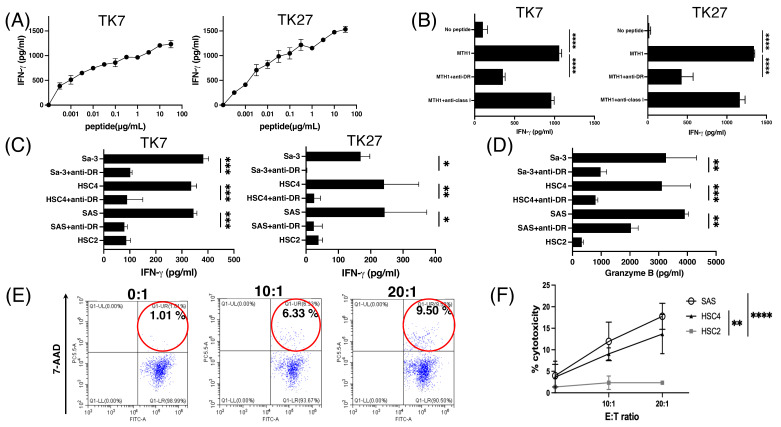
Induction of MTH1_45–59_ peptide-specific CD4+ T-cells. (**A**) Evaluation of IFN-γ production by MTH1_45–59_-reactive HTLs (TK7 and TK27) with peptide stimulation. These T cells were co-cultured with autologous PBMCs as antigen-presenting cells (APCs) with various MTH1 peptide concentrations. (**B**) HLA restriction of MTH1_45–59_-reactive HTLs. IFN-γ production from T cells with MTH1_45–59_ peptide-pulsed autologous PBMCs was evaluated in the presence of an anti-HLA-DR Ab or anti-HLA class I Ab (negative control). The production of IFN-γ in supernatants was measured by ELISA after coculturing for 48 h. (**C**) Direct recognition of tumors by MTH1_45–59_-reactive CD4+ T cells was evaluated by co-culturing HLA-DR-matched or-unmatched tumor cell lines. The production of IFN-γ in supernatants was measured by ELISA after coculturing for 48 h. TK7 and TK27 were derived from a donor with HLA-DR 9/12/53. SAS: HLA-DR9/15/53; HSC4: HLA-DR1/4/53; Sa-3: HLA-DR9/10/53. HSC2 (HLA-DR13/13) was used as an HLA-DR-unmatched negative control. All tumor cells were treated with IFN-γ (50 IU/mL) for 24 h before the assay. (**D**) Granzyme-B production from MTH1_45–59_-reactive CD4+ T-cell line was assessed by co-culturing HLA-DR-matched or HLA-DR-unmatched HNSCC cell lines. HSC2 (HLA-DR13/13) was used as an HLA-DR-unmatched negative control. Each data point was representative of triplicate experiments. Bars and error bars represent mean and SD, respectively. (* *p* < 0.05, ** *p* < 0.01, *** *p* < 0.001, **** *p* < 0.0001, Student’s *t*-test). (**E**,**F**) Killing activity of MTH1-reactive HTLs in tumor cell lines with several effector-to-target (E: T) ratios. HLA-DR53-restricted TK7 cell lines were co-cultured with tumor cell lines at several effector-to-target (E: T) ratios. Tumor cells with irrespective HLA-DR (HSC2: HLA-DR13) were used as a negative control. MTH1-reactive HTLs were co-cultured with CFSE-labeled tumor cells for 6 h, and dead cells were labeled with 7-AAD. The percentage of dead tumor cells (CFSE+ 7-AAD+ cells) was determined using flow cytometry.

**Figure 5 cancers-16-03013-f005:**
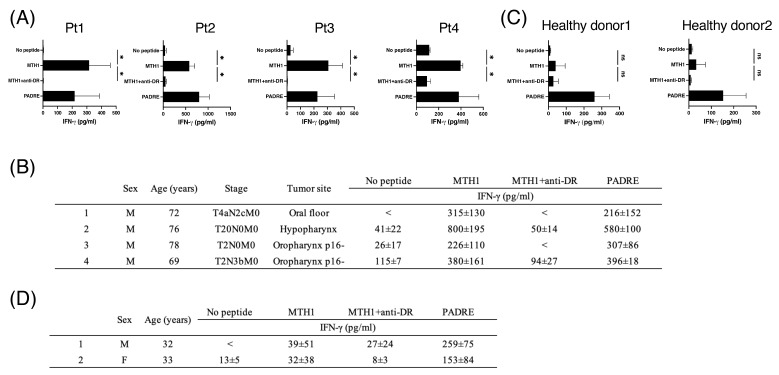
The existence of MTH1-reactive precursor T cells in HNSCC patients. (**A**,**C**) PBMCs from (**A**) patients with HNSCC or (**C**) healthy donors were stimulated with MTH1_45–59_ peptides for 2 cycles every week. T-cell response to the MTH1_45–59_ peptide was evaluated by measuring IFN-γ in the supernatants. Anti-HLA-DR mAb was used to assess HLA-DR restriction, and the PADRE peptide was used as a positive control. Each data point was representative of triplicate experiments. Bars and error bars represent mean and SD, respectively. (* *p* < 0.05, one-way ANOVA). (**B**) The clinical characteristics and peptide reactivities of the patients with HNSCC. <: less than lower limit of detection. (**D**) The clinical characteristics and peptide reactivities of the 2 healthy donors (HLA-DR12/14 and -DR4/15/53). <: less than lower limit of detection.

**Figure 6 cancers-16-03013-f006:**
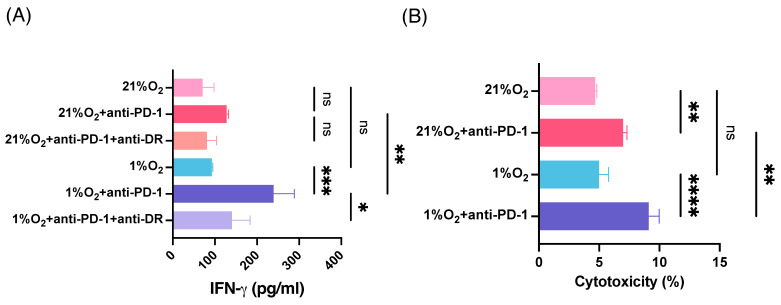
The cytotoxicity of MTH1-reactive CD4+ T-cells against HNSCC cells with PD-1 blockade in hypoxic conditions. (**A**) Direct recognition of tumors by MTH1_45–59_-reactive CD4+ T cells was evaluated by co-culturing HLA-DR-matched tumor cell line (SAS) and measuring IFN-γ production with or without the anti-PD-1 antibody under normoxic and hypoxic conditions. The production of IFN-γ in supernatants was measured by ELISA after coculturing for 48 h. (**B**) Killing activity of MTH1-reactive HTLs in HLA-DR-matched tumor (SAS) with or without the anti-PD-1 antibody under normoxic or hypoxic conditions. The ratio of T cells to tumor was 20:1. Tumor cells were treated with IFN-γ (50 IU/mL) for 24 h before the assay. Bars and error bars represent mean and SD, respectively. (* *p* < 0.05, ** *p* < 0.01, *** *p* < 0.001, **** *p* < 0.0001, one-way ANOVA).

**Figure 7 cancers-16-03013-f007:**
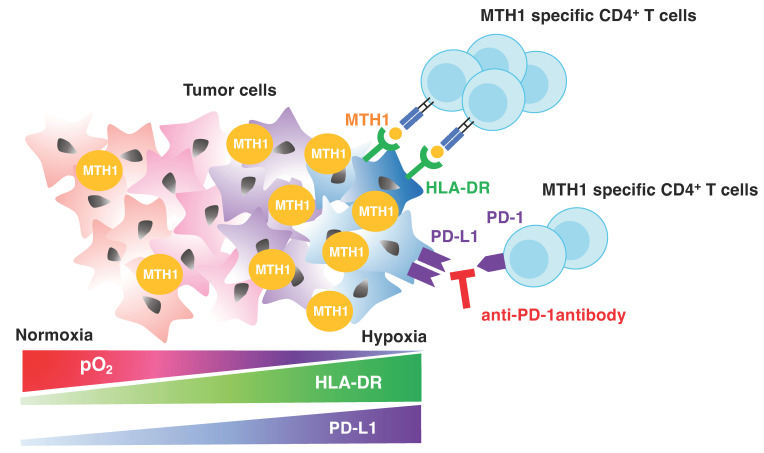
MTH1-targeted CD4-based antitumor immunotherapy with anti-PD-1 antibodies. The expression of MTH1, PD-L1, and HLA-DR in tumor cells is upregulated under hypoxic conditions. The MTH1 peptide vaccine or adoptive cell transfer of MTH1-specific CD4+ T cells in combination with PD-1 blockade would be a promising strategy to treat tumor in hypoxic conditions.

**Table 1 cancers-16-03013-t001:** Relationship between clinicopathologic features of patients with oropharyngeal cancer and MTH1 expression.

Clinical Feature	MTH1 IHC Score		*p* Value
	Low	High	
Gender			
Female	1	6	0.2187
Male	22	26	
Age (years)			
<70	14	17	0.5678
≥70	9	15	
HPV status			
Negative	11	19	0.3962
Positive	12	13	
T classificaion			
T1–2	11	18	0.5371
T3–4	12	14	
N classification			
N0–1	16	20	0.5868
N2–3	7	12	
Stage			
I, II	13	13	0.2441
III, IV	10	19	

IHC, immunohistochemistry; HPV, human papillomavirus.

## Data Availability

The data that support the findings of this study are available on request from the corresponding author. The data are not publicly available due to privacy or ethical restrictions.

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
