# Peer review of "Hypoxia-Targeted Immunotherapy with PD-1 Blockade in Head and Neck Cancer"

_cancers, 2024, doi:10.3390/cancers16173013_

Round 1
Reviewer 1 Report
Comments and Suggestions for Authors
The authors investigated the immune dynamics of tumor cells and T cells under a hypoxia environment. They confirmed the presence of cytotoxic MTH1-specific CD4+T cells, and demonstrated the possibility of enhancing cytotoxic activity by anti-PD-1 antibody. The research focus is interesting, and the manuscript is well-written.
1. Are there differences in T cell proliferation and cell cycle proportion between normoxic and hypoxic conditions?
2. What is the estimated oxygen pressure in tumor?
3. Although MTH-reactive precursor T cells could not be detected in PBMC from healthy donors, the authors describe that precursors of MTH1-reactive HTLs were identified in a healthy donor (line 373-374). Isn’t this a contradiction?
4. Is MTH1 expressed in cells other than tumor cells? Is its expression increased in hypoxic conditions? If so, what about the possibility that cells other than tumor cells being lysed?
Author Response
Thank you for reviewing and considering our manuscript interesting. We appreciate all the comments from this reviewer, which have further improved the merits of this paper.
- Are there differences in T cell proliferation and cell cycle proportion between normoxic and hypoxic conditions?
>Answer: We thank the reviewer for pointing out an important issue. As shown in Supplementary Figure 1, the proliferation of CD8 T cells was slightly decreased with hypoxia. Although CD4 T cells in the G2/M phase were increased in normoxia compared to hypoxia, the proliferation of CD4 T cells was comparable regardless of low oxide concentration and these cells were capable of killing tumor cells. We have added this information in the revised manuscript (Line 208-210).
- What is the estimated oxygen pressure in tumor?
>Answer: A hypoxic environment exists in solid tumors owing to an imbalance between oxygen supply and demand. Oxygen concentrations in tumors have been reported to range from 1% to 2% (PMID: 24588669, 27774485). Accordingly, we chose 1% as a condition that represents hypoxic tumor. We have added this information in the revised manuscript (Line 175-178).
- Although MTH-reactive precursor T cells could not be detected in PBMC from healthy donors, the authors describe that precursors of MTH1-reactive HTLs were identified in a healthy donor (line 373-374). Isn’t this a contradiction?
>Answer: We apologize for the confusion. To detect precursor T cells, PBMCs from patients or donors were activated with peptide only for two weeks as shown in the Materials and Methods due to the small volume of available blood samples. To elicit MTH1-reactive HTLs, CD4 T cells purified from PBMCs were educated by professional antigen presenting cells (dendritic cells), and stimulated multiple times over 4-cycles every 7 days. Thus, we believe that the few precursors of these T cells exist in healthy donors, but would be difficult to detect with the short-term activation protocol. These cells should be stimulated with professional APCs for several times to be detected. The precursor T cells are increased in the patients with cancer that contains tumor antigen (MTH1), and could be detected with the short-term activation protocol. We have added this information in the revised manuscript (Line 372-374).
- Is MTH1 expressed in cells other than tumor cells? Is its expression increased in hypoxic conditions? If so, what about the possibility that cells other than tumor cells being lysed?
>Answer: We thank the reviewer for raising this important issue. Normal tissue expresses MTH1 but its expression is low. We have performed additional experiments using normal human bronchial epithelial cells (NHBE) and found that the expression of MTH1 was increased in tumor compared to NHBE (Supplementary Figure 3A). In immunohistochemical staining, MTH1 was highly expressed in head and neck cancer compared to surrounding normal tissue (Supplementary Figure 3B). This aberrant expression of MTH1 in tumor compared to normal tissue is also confirmed in lung, colon, and pancreatic tissues (PMID: 31311767). Because the MTH1 dependency is low in normal tissue compared to tumor (PMID: 33354500, 28447629), hypoxia did not increase MTH1 expression in NHBE (Supplementary Figure 3C). It should be noted that the patients with MTH1-reactive precursor T cells have shown no autoimmune diseases. Thus, we believe that MTH1-reactive T cells that may have low affinity T cell receptor only react to MTH1-high tumor but ignore MTH1-low normal tissues. We have added this issue in the revised manuscript (Line 234-238, 376-378).
Reviewer 2 Report
Comments and Suggestions for Authors
This is a novel and a detailed study about MTH1 in the context of cancer. The study is detailed and well executed. However, the authors discuss about MTH1 only in their Discussion section. They speak about nitric oxide signaling under hypoxia but must also mention that low nitric oxide under hypoxic conditions also promotes vasodilation (10.1089/ars.2012.4979). Under hypoxia, IDO1 is overexpressed (10.18632/oncotarget.24098) which is a key promoter of cancer due to immune suppression (10.3389/fimmu.2021.636081). Nitric oxide signaling under hypoxia also promotes heme insertion and activation of IDO1 (10.1016/j.jbc.2023.104753). The authors should discuss these papers in their Discussion section. Thank you.
Author Response
This is a novel and a detailed study about MTH1 in the context of cancer. The study is detailed and well executed. However, the authors discuss about MTH1 only in their Discussion section. They speak about nitric oxide signaling under hypoxia but must also mention that low nitric oxide under hypoxic conditions also promotes vasodilation (10.1089/ars.2012.4979). Under hypoxia, IDO1 is overexpressed (10.18632/oncotarget.24098) which is a key promoter of cancer due to immune suppression (10.3389/fimmu.2021.636081). Nitric oxide signaling under hypoxia also promotes heme insertion and activation of IDO1 (10.1016/j.jbc.2023.104753). The authors should discuss these papers in their Discussion section. Thank you.
>Answer: We thank this reviewer for reviewing our paper and providing valuable advice.
We agree that hypoxia could elicit immune suppression through IDO1, whereas vasodilation by nitric oxide may increase the access of immune cells to tumor microenvironment. We have added these references and discussion in the revised manuscript (Line 413-417).
Reviewer 3 Report
Comments and Suggestions for Authors
I found this translational research article to be of particular interest.
From a methodological standpoint, the study is sound, and the clinical perspectives are noteworthy.
From my perspective, the article is publishable in its current form. To enhance comprehensiveness, I would request the authors to provide insights on potential synergies with systemic and radiotherapeutic treatments. It is widely acknowledged that radiotherapy does not directly target hypoxic tissues, suggesting the possibility of identifying favourable combinations. This represents a promising avenue for future investigation.
Comments on the Quality of English LanguageI have no particular comments.
Author Response
I found this translational research article to be of particular interest.
From a methodological standpoint, the study is sound, and the clinical perspectives are noteworthy.
From my perspective, the article is publishable in its current form. To enhance comprehensiveness, I would request the authors to provide insights on potential synergies with systemic and radiotherapeutic treatments. It is widely acknowledged that radiotherapy does not directly target hypoxic tissues, suggesting the possibility of identifying favourable combinations. This represents a promising avenue for future investigation.
>Answer: We thank the reviewer for the valuable information that enhances the significance of this study. We agree with the reviewer that the combination of radiotherapy and MTH1-targeted immunotherapy would be promising, and have added this information in the revised manuscript (Line 420-424).